# Effects of different vegetable rotations on the rhizosphere bacterial community and tomato growth in a continuous tomato cropping substrate

Li Jin[1], Jian Lyu[1,2]*, Ning Jin[1], Jianming Xie[1], Yue Wu[1], Guobin Zhang[1], Zhi Feng[1], Zhongqi Tang[1], Zeci Liu[1], Shilei Luo[1], Jihua Yu[1,2]*

1 College of Horticulture, Gansu Agricultural University, Lanzhou, China, 2 Key Laboratory of Crop Science in arid environment of Gansu Province, Lanzhou, China

* lvjiangs@126.com (JL); yujihuagg@163.com (JY)

**Data Availability Statement:** All relevant data are within the paper and its Supporting information files and uploaded to the NCBI Sequence Read

## Abstract

In this study, High throughput sequencing was used to analyze the effects of different vegetable rotations on the rhizosphere bacterial diversity and community structure in a substrate that was used for continuous tomato cropping (CK). The vegetable rotations tested were cabbage/tomato (B), kidney bean/tomato (D), and celery/tomato (Q). The results revealed that the substrate bacterial diversity and richness of each crop rotation were higher than those of CK. The highest bacterial diversity was found in the B substrate, followed by the Q and D substrates. Further comparison showed that the rhizosphere bacterial community structure of Q substrate was significantly different to that of CK. Compared with the CK, the Q substrate had a significantly higher relative abundance of several dominant microflora, such as Acidobacteria, Chloroflexi, and Firmicutes. Additionally, the Q rotation significantly increased the abundance of beneficial bacteria, such as *Actinobacteria_unclassified* and *Anaerolineaceae_unclassified*. A redundancy analysis showed that Most dominant bacteria correlated positively with the substrate pH, total N, and alkali-hydrolyzable N but negatively with the available P, available K, total P, total K, and organic matter contents and substrate EC. The substrates after crop rotation improved the growth and physiological condition of the subsequent tomato plants, among which those from the Q rotation performed the best. Therefore, celery rotation not only increased the richness and diversity of bacterial communities in the substrate but also significantly increased the richness of the beneficial bacterial communities, allowing better maintenance of the substrate microenvironment for the healthy growth of crops.

## Introduction

Organic substrates are widely applied for plant cultivation in solar greenhouses in China, having been easily extended owing to their low costs, less strict requirements of the external

Archive (SRP253823) (https://identifiers.org/ncbi/insdc.sra:SRP253823).

**Funding:** This research was supported by Special project of central government guiding local science and technology development (ZCYD-2020-5-2), National key R & D plan (2018YFD0201205), Special project of national modern agricultural industrial system (CARS-23-C-07), Gansu people's livelihood science and technology project (20CX9NA099).

**Competing interests:** The authors have declared that no competing interests exist.

environment, and simple cultivation management. However, the practice of continuous cropping eventually leads to deterioration of the physicochemical properties of the organic substrate, and aggravation of plant diseases and insect pests [1]. Aside from the deterioration of the substrate, continuous monoculture can also increase the presence of harmful microorganisms, thereby destroying the microecological balance [2]. Studies have shown that substrate problems still lead to continuous cropping obstacles, and the high temperature and humidity in the greenhouse serve to aggravate the occurrence of crop diseases and insect pests. Therefore, the ability to reuse the substrate is the main problem restricting the development of soilless substrate cultivation [3].

Because of its high potential yield, crop water productivity, and profitability, the tomato plant (*Lycopersicon esculentum* Mill., belonging to the family Solanaceae) has quickly become one of the major vegetables grown in solar greenhouses in arid regions of northwestern China [4]. With the increase in tomato planting areas in the region, the continuous cultivation of the crop is becoming increasingly prevalent [5]. It is well known that the continuous cropping of a single species causes plant growth retardation, serious diseases and insect pests, low crop productivity, soil degradation, and microecosystem imbalance [6,7]. Compared with monoculture, Crop rotation (e.g., leguminous crop diversification) is a strategy used for maintaining soil quality and crop productivity to the high degree that single cropping cannot [8]. Horticultural/herbaceous crop rotation affects the physical and chemical properties of soil and its microbial community through the difference in the quantity and composition of root exudates, so as to improve the soil biological environment, reduce continuous cropping obstacles, and improve the yield of horticultural crops. Moreover, rotation can maintain soil properties within an acceptable range and achieve the goal of sustainable agriculture [9–12]. Additionally, the different plant–microbe interactions also affect the soil microbial communities, leading to less soil-borne diseases [13,14]. The diversity and richness of soil microorganisms play a particularly important role in maintaining soil health and promoting crop growth [15]. In recent years, most studies on ways to overcome or alleviate the obstacles of continuous cropping have focused mainly on changes in the rhizosphere soil microorganisms after the continuous cultivation of crops [16,17]. However, there are few reports on the changes in the organic substrate quality and microorganisms effect by tomato monoculture for several years [18].

In this study, a high-throughput sequencing method was used to study the bacterial diversity and community structures in organic substrates that had first been used for continuous tomato cropping and then rotated for the cultivation of different vegetables (cabbage, kidney bean, and celery). We compared the continuous tomato cropping system with three different vegetable/tomato rotation systems to achieve the following objectives: (1) to determine the changes in the bacterial diversity and community structure in the continuous tomato cropping substrate; (2) to explore the relationship between the bacterial community and physicochemical characteristics of the substrate; and (3) to evaluate which vegetable is the best for tomato rotation.

## Materials and methods

### Experimental design

The materials and methods are similar to those of "Lyu" [2]:

The test substrate is an organic ecotype soilless culture substrate derived from local agricultural waste, which is made up of a mixture of slag, spent mushroom, cow manure, chicken manure, and corn straw at the ratios of 13:5:5:2:14, respectively. It is mainly suitable for tomato substrate cultivation and has been vigorously promoted locally [19]. From June 2012 to June 2018, the continuous cropping experiment was conducted in the solar greenhouse of

"Zongzhai non cultivated land facility agricultural demonstration park" in Zongzhai Town, Suzhou District, Jiuquan City, Gansu Province, China (98° 20′ ~ 99° 18′ E, 39° 10′ ~ 39° 59′ N; a region with a typical continental climate, with an average sea level of 1360 m, annual average temperature of 7.3°C, annual average precipitation of 176 mm, and annual sunshine hours of 3033–3316 h). The continuous cropping vegetable was tomato (*Lycopersicon esculentum* Mill.), which was planted twice a year. In China, the overwintering crop is generally raised in September, planted in October, collected in the first 10 days of February of the next year, and pulled on time in May of that same year. Summer and autumn crops are generally planted according to the overwintering schedule, with most being planted in June, harvested in the first 10 days of August, and planted on time again in October. The experimental tomato variety used was "Jingfan 501," a pink fruit of infinite growth type. The plants were spaced 45 cm apart, with a row spacing of 25 cm and 30 plants in each plot (3 replicates). After continuous cropping (with 12 crops planted over 6 years), the substrate (designated CK) would usually contain the following amounts of macronutrients: total potassium (K) of 11.78 g·kg$^{-1}$, total phosphorus (P) of 1.31 g·kg$^{-1}$, total nitrogen (N) of 0.51 g·kg$^{-1}$, available P of 82.81 mg·kg$^{-1}$, available K of 63.17 mg·kg$^{-1}$, and alkali-hydrolyzable N of 907.67 mg·kg$^{-1}$. Moreover, it typically has an electrical conductivity (EC) of 1683.67 μS·cm$^{-1}$ and a pH of 6.37.

The crop rotation experiments were conducted in a glass greenhouse at Gansu Agricultural University from August 2018 to March 2019. The rotation vegetables were cabbage (*Brassica pekinensis* Rupr. Jingdong216), kidney bean (*Phaseolus vulgaris* Linn. Hongshuai), and celery (*Apium graveolens* L. Jinhuanghou), designated B, D, and Q, respectively, and tomato continuous cropping was designated CK. The CK substrate collected from the continuous tomato cropping experimental site was transported to the greenhouse at Gansu Agricultural University and placed into 19 cm × 30 cm pots (5 kg of substrate for each pot). The various vegetable seedlings that had been raised in advance were individually subplanted into pots. Four cabbage plants are planted in each pot, a small amount of water is watered for many times in the early stage, and watering and fertilization are applied once a week in the peak growth period. The growth cycle of cabbage is short, so a total of two crops are planted. Two kidney beans are planted in each pot, and the vine is hung during vine extension. Fertilization and irrigation are carried out every two weeks in the early growth stage, and once every week in the peak pod setting stage. Two celery plants are planted in each pot. After planting, a small amount of water is watered for many times to promote the growth of seedlings. There is less water in the early stage of growth to prevent overgrowth. When entering the period of vigorous nutritional growth, water and fertilizer are applied once a week. Other field management measures were consistent with local conventional management measures. Each rotation experiment was carried out in triplicate.

## Substrate sampling

The substrate samples were collected after the rotation plant had been pulled (March 2019). After removing the 0–5 cm surface substrate and gently shaking off the loose substrate around the root system, the substrate adhered to the root surface was brushed off for collection and immediately stored as the rhizosphere sample in an ice box. The rhizosphere samples were extracted in triplicate for each the four crop experiments (i.e., CK, B, D, and Q), totaling 12 samples.

## DNA extraction

The total DNAs of the microorganisms in 0.5 g of rhizosphere sample were extracted using the EZNA® Soil DNA Kit (OMEGA, Bio-Tek, Norcross, GA, USA) according to the

manufacturer's protocols. For each crop experiment, the solutions of DNA extracted from the triplicate samples were pooled.

## Illumina MiSeq sequencing

The purified DNA was used as a template for polymerase chain reaction (PCR) amplification of the 16S rDNA V3+V4 region, using the primers 338f (5′- ACTCCTACGGGAGGCAGCAG-3′) and 806r (5′-GGACTACHVGGGTWTCTAAT-3′) [20]. The 25-μL reaction system consisted of 12.5 μL of Phusion Hot Start Flex 2× Master Mix, 2.5 μL of forward primer, 2.5 μL of reverse primer, 50 ng of template DNA, and enough ddH$_2$O to make up the final volume. The PCR was carried out on an ABI GeneAmp® system (Model 9700; Applied Biosystems, Foster City, CA, USA) with the following amplification conditions: predenaturation at 98˚C for 30 s, denaturation at 98˚C for 10 s, annealing at 54˚C for 30 s, extension at 72˚C for 45 s, for a total of 35 cycles, and a final extension at 72˚C for 10 min. Each sample was prepared in triplicate, and the final amplification product was detected by 1% agarose gel electrophoresis. The PCR products were then purified using an AxyPrep DNA gel extraction kit (AxyGen Biosciences, Union City, CA, USA) and quantified using a Quant-iT PicoGreen dsDNA assay kit (Promega, Madison, WI, USA) on a quantitative fluorescence system (QuantiFluor, Promega). (The qualifying library concentration should be more than 2 nM.) After diluting the qualified online sequencing libraries (the index sequence is not repeatable), they were mixed to the required sequencing amount according to the corresponding proportion. NaOH was then used to transform the molecules into a single chain for online sequencing with the MiSeq sequencer. For 2 × 250-bp double-terminal sequencing, the corresponding reagent to use is MiSeq Reagent Kit v2500 cycles.

## Processing of the sequencing data

The original MiSeq data were obtained as an image file. After base calling, a result file stored in fastq format was obtained and then quality filtered using QIIME (version 1.17). PEAR (version 0.9.6) [21] and Vsearch (version 2.3.4) [22] were used to splice the two terminal sequences of the original data and filter the chimeric sequence. The Vsearch algorithm clustered the sequences with a greater than 97% similarity. Then, the representative operational taxonomic unit (OTU) sequences obtained by the clustering analysis were compared with sequences on the Ribosomal Database Project (version 11.5) [23,24] and SILVA databases [25] to obtain the species annotation results of all OTUs.

## Determination of the growth physiology of subsequent tomato plants

**Growth index.**   The plant height was determined as the height from the base of the root to the growth point of the plant. The leaf area was determined using an LI-3000C leaf area meter (LI-COR, USA).

**Physiological indices.**   Photosynthetic parameters: From tomato plants at their peak, the leaves at the same leaf position (i.e., the second leaf under the first inflorescence) were selected to measure their net photosynthetic rate (Pn), transpiration rate (Tr), stomatal conductance (Gs), and intercellular CO$_2$ concentration (Ci), using a CIRAS-2 (PP-System, UK) portable photosynthetic analyzer on the morning of a clear day. The values for each leaf were read three times.

**Fluorescence parameters.**   Well-growing functional leaves were selected for determination of the levels of initial fluorescence (Fo), maximum fluorescence (Fm), variable fluorescence (Fv), photochemical quenching of photosystem II (q$^P$), and non-photochemical

quenching (NPQ), using an FMS-2 portable pulse-modulated fluorescence meter (Hansatech Instruments, Norfolk, UK).

## Statistical analysis

Statistical analysis of the data was performed using the R packages Stats and Vegan (version 2.3–5) [26]. Alpha-diversity indices (Shannon index, Simpson index, Chao1 index, and number of observed species) were calculated using QIIME (alpha_diversity.py). For the beta-diversity analysis, cluster analysis (sample clustering by Bray-Curtis distance) was used to show the similarity between samples, and weighted UniFrac distance measurement (based on system development structure) was used to generate a principal coordinate analysis (PCoA) map to further evaluate the similarity between community members of the samples [27]. Redundancy analysis (RDA) was performed with the RDA function in the Vegan package in R (version 2.1.3) to study the effects of the physicochemical properties of the substrate on the species abundance of the rhizosphere bacterial community. The tomato growth physiological data were analyzed by one-way analysis of variance using Excel 2016 and SPSS 20.0. The significance level for differences in this study was set at $P < 0.05$.

## Results

### Effects of different vegetable rotations on the bacterial diversity in the continuous tomato cropping substrate

The changes in bacterial alpha-diversity index values of the various rotation and continuous cropping rhizosphere samples are shown in Table 1. The number of observed species (3242.33) and Chao1 index (6031.56) were the highest for the B rhizosphere, indicating that it had the highest bacterial community diversity and richness. The D rhizosphere had the highest Shannon index (10.58). Additionally, compared with CK, the 3 vegetable rotations showed an increasing trend for the observed species, Shannon index and Chao1 index of bacteria. These results showed that all four cropping systems tested could improve the richness and diversity of bacteria in the CK substrate to a certain extent.

Fig 1A shows the beta-diversity of the bacterial communities in the each crop experiment substrates as obtained by hierarchical clustering analysis. As shown in the figure, the bacterial communities were divided into three cluster groups: groups CK and D, group Q, and group B. These groupings indicated that the bacterial community structures of the CK and D rhizospheres were similar but vastly different from those of the other two rhizospheres, and the bacterial community structures of the Q and B rhizospheres were also different from each other. Additionally, the difference in the fungal community composition between the B and CK rhizospheres was the greatest, and B was separated from those of the D and Q rhizospheres.

**Table 1. Comparison of species numbers and 16S (bacterial) diversity indices observed in the different cropping systems.**

| Cropping system | Observed species | Shannon index | Simpson index | Chao1 index |
|---|---|---|---|---|
| CK | 2839.33 ± 112.34b | 10.40 ± 0.07a | 1.00 ± 0.00a | 5129.79 ± 364.28a |
| Q | 2978.67 ± 25.12ab | 10.46 ± 0.02a | 1.00 ± 0.00a | 5472.59 ± 223.76a |
| B | 3242.33 ± 122.83a | 10.51 ± 0.09a | 1.00 ± 0.00a | 6031.56 ± 339.80a |
| D | 3051 ± 133.63ab | 10.58 ± 0.05a | 1.00 ± 0.00a | 5373.15 ± 449.44a |

CK: Continuous tomato cropping; Q: Celery/tomato rotation; B: Cabbage/tomato rotation; D: Kidney bean/tomato rotation. Different lowercase letters in a row indicate that the differences are statistically significant (P < 0.05).

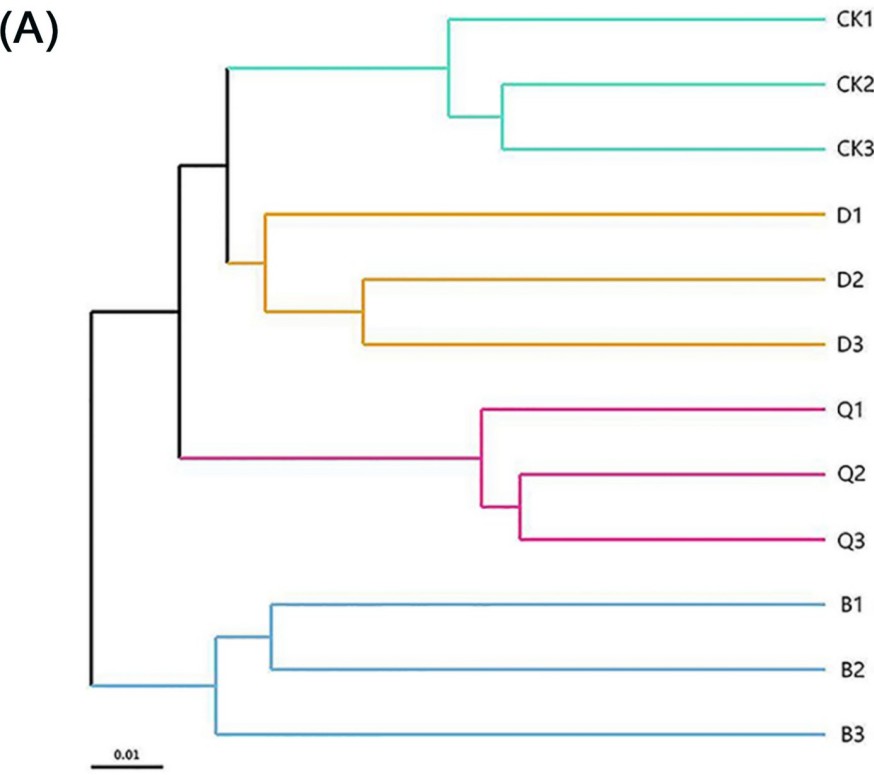

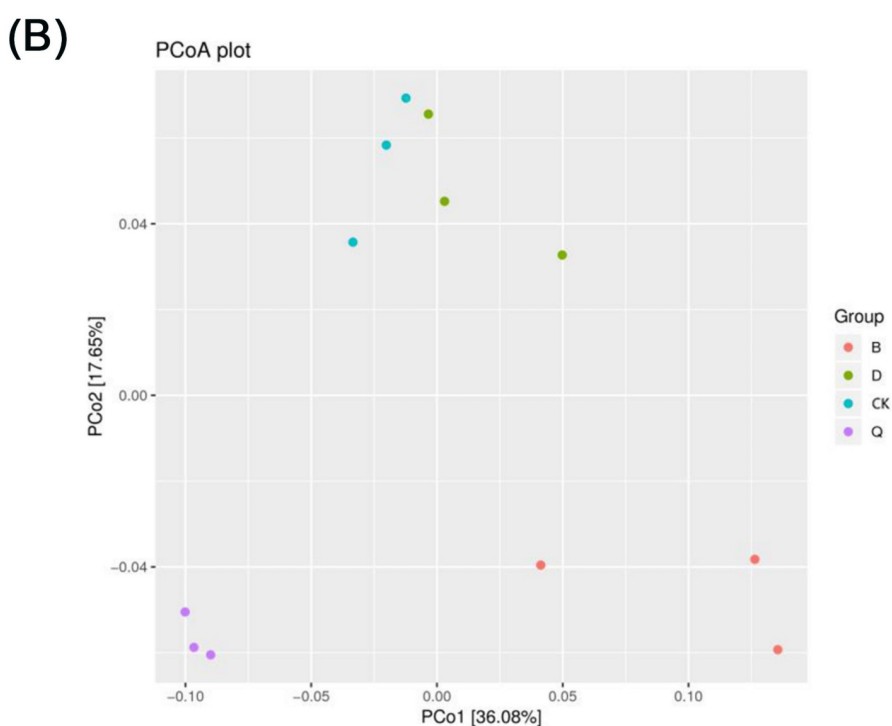

**Fig 1. Hierarchical clustering (A) and principal coordinate analysis (B) of the bacterial communities in the continuous tomato cropping substrates based on different cropping systems.** CK: Continuous tomato cropping; B: Cabbage/tomato rotation; D: Kidney bean/tomato rotation; Q: Celery/tomato rotation.

The UniFrac weighted PCoA based on OTU components also clearly showed the difference between the final CK substrate and those from the different vegetable rotations, where the two axes explained 36.08% and 17.65% of the total bacterial data, respectively. Additionally, the bacterial communities in the CK and D rhizosphere samples ordinated closely together in PCo2 and were significantly separated from those of the B and Q rhizospheres. (Fig 1B).

## Effects of different vegetable rotations on the bacterial community structure in the continuous tomato cropping substrate

Although the diversity of the bacterial community structures in the four samples was similar, the abundance was different. The abundance of the bacterial community structure under each crop experiment is shown at the phylum level in Fig 2. In total, 32 bacterial phyla were observed. The common taxa in the substrate were Proteobacteria (37.04%–34.09%), Actinobacteria (20.75%–19.68%), Acidobacteria (13.54%–6.97%), Bacteria_unclassified (7.75%–7.02%), Chloroflexi (8.98%–6.29%), Gemmatimonadetes (7.29%–5.17%), and Firmicutes (4.30%–3.44%). The abundance of Proteobacteria was the highest in the B rhizosphere, accounting for 37.04% of the total bacteria in that substrate, and was 8.65% higher than that in the control CK substrate, which had the lowest Proteobacteria richness. The abundance of Actinobacteria was the highest in the CK substrate, whereas it was decreased in all the rotation substrates, being the lowest in the B rhizosphere (5.16% lower than that of CK). The Q rhizosphere had the highest abundance of Chloroflexi and Firmicutes, which were 24.55% and 20.79% higher than that in the CK rhizosphere, respectively.

Fig 3 shows the thermogram of the diversity and richness of the bacterial communities in the four different rhizosphere samples, as represented by color gradient and similarity, where the color change from blue to red indicates a change in relative abundance of the community from low to high. In each crop rhizosphere, the proportion of *Bacteria_unclassified* (7.75%–7.02%) was the highest, with the order according to crop experiment being Q > B > CK > D. The genus in the second highest proportion was *Gemmatimonas* (7.28%–5.16%), in the order CK > B > D > Q. *Sphingomonas* and *Gemmatimonas* were the main genera in the substrates

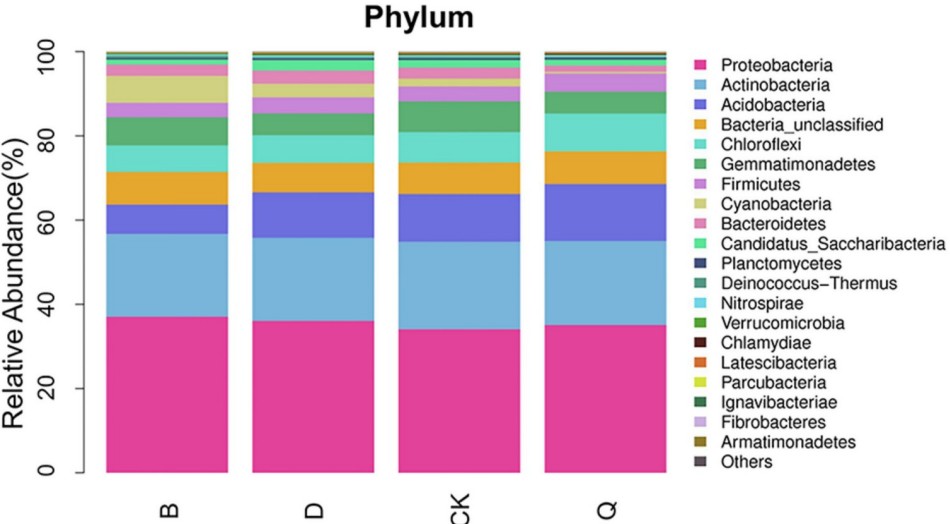

**Fig 2. Relative abundance of main bacterial genera (top 20) in continuous tomato cropping substrates under different cropping systems.** CK: Continuous tomato cropping; B: Cabbage/tomato rotation; D: Kidney bean/tomato rotation; Q: Celery/tomato rotation.

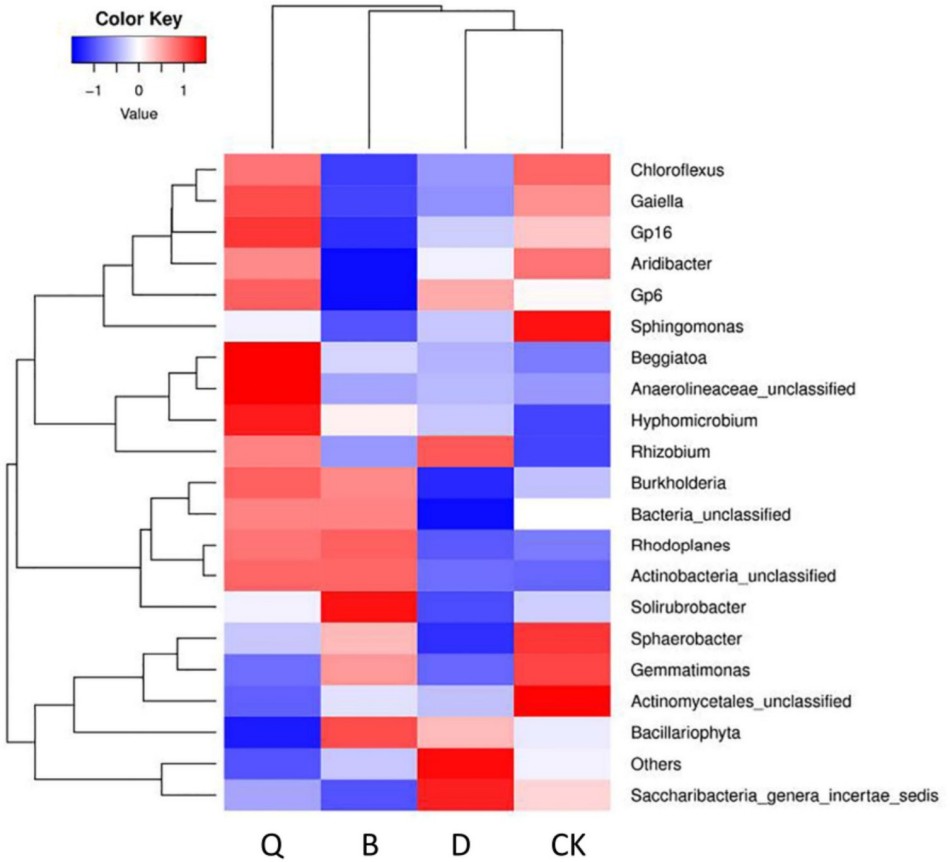

**Fig 3. Heatmap of the changes in bacterial community richness in the continuous tomato cropping substrates under different cropping systems.** CK: Continuous tomato cropping; Q: Celery/tomato rotation; B: Cabbage/tomato rotation; D: Kidney bean/tomato rotation.

for continuous cropping, at abundance values of 1.41% and 7.28%, respectively. In the Q rhizosphere samples, *Hyphomicrobium*, Anaerolineaceae_unclassified, Actinobacteria_unclassified, and *Chloroflexus* occurred at an abundance of 2.08%, 2.27%, 2.51%, and 1.73%, respectively. In the B rhizosphere, *Solirubrobacter* and *Bacillariophyta* were dominant, occurring at an abundance of 2.36% and 5.86%, respectively. In the D sample, *Rhizobium* and *Saccharibacteria_ genera_ incertae_sedis* were dominant, being present at an abundance of 1.64% and 2.45%, respectively.

## Effects of the different vegetable rotations on tomato growth

The height, stem diameter, and leaf area of the tomato plants grown under the different cropping systems were similar; that is, they all increased with the growth and development of the plants (Fig 4). At the beginning of flowering, the height of the Q plant was the highest(0.44%), being significantly higher than that of the CK plant. By contrast, there was no significant difference in height between the B, D, and CK plants at this stage. At the early fruiting stage, the D plant was significantly taller (0.46%) than the CK plant. By contrast, there was no significant difference in height between the B and Q plants. With the extension of the growth period, the height of the B plant increased the most obviously and reached its maximum value in the last

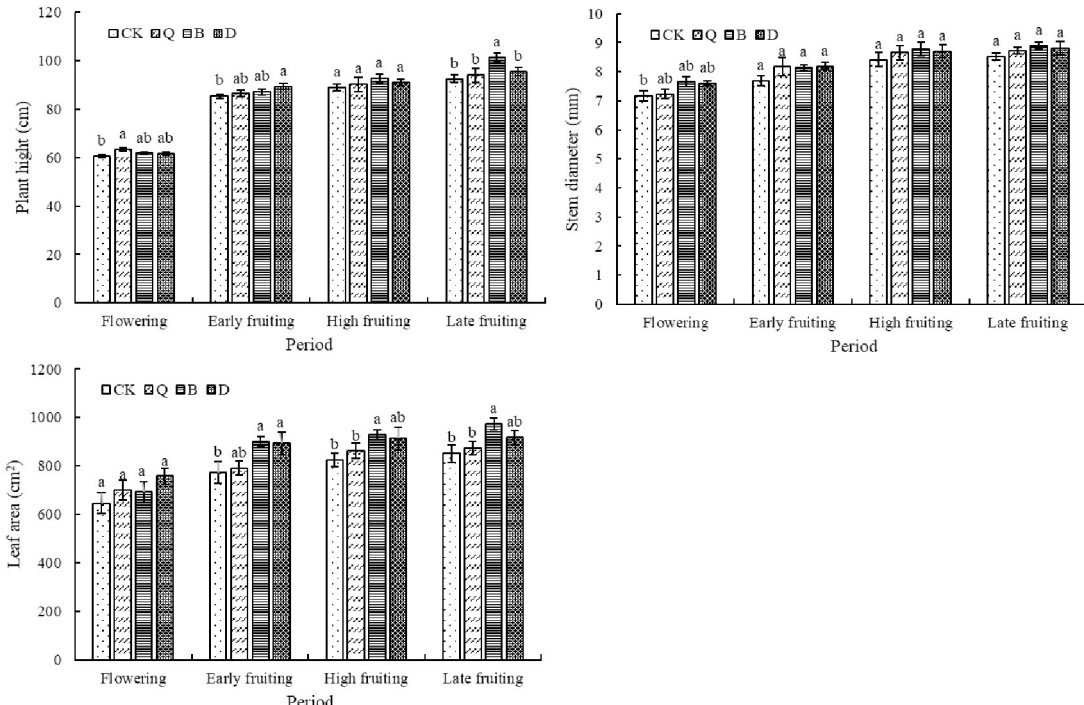

**Fig 4. Effects of different vegetable rotations on tomato plant growth.** CK: Continuous tomato cropping; Q: Celery/tomato rotation; B: Cabbage/tomato rotation; D: Kidney bean/tomato rotation. Different lowercase letters at each phenological stage indicate that the differences are statistically significant (P < 0.05).

two periods. Although there was no significant difference in stem diameter among the plants from all four cropping systems during the growth period, the value for each vegetable rotation was higher than that for CK. There was also no significant difference in the leaf area among the plants from all four cropping systems at the beginning of flowering, with that of the B plant being the smallest. With the extension of the growth period, the leaf area of the B plant increased gradually, reaching its maximum in each subsequent growth period. The leaf area of the D plant was significantly larger (15.42%) than that of the CK plant at the early fruiting stage, but the difference between the two plants was no longer significant at the peak and late fruiting stages. The leaf area of the Q plant was not significantly different from that of the CK plant, albeit it was still larger.

### Effects of the different vegetable rotations on photosynthetic parameters of the tomato leaves

The effects of different vegetable rotations on the light and photosynthetic parameters of the tomato leaves are shown in Table 2. Overall, the light and photosynthetic parameters of the leaves from tomato plants grown in the three crop rotation systems were higher than those of the leaves from the CK system. The Ci of the Q leaves was significantly higher (38.15%) than that of the CK leaves. The Tr and GS of the B and D leaves were significantly higher than those of the CK leaves (Tr increases of 62.99% (Q), 69.12% (B), and 58.86% (D) and Gs increases of 65.06% (Q), 50.85% (B), and 62.81% (D), respectively). Compared with the Pn of the CK leaves, that of the Q leaves was significantly higher (by 48.46%), whereas there was no significant difference between the B, D, and CK leaves for this parameter.

**Table 2. Effects of different vegetable rotations on light and photosynthetic parameters of the tomato leaves.**

| Cropping system | Ci | Tr | Gs | Pn |
|---|---|---|---|---|
| | ($\mu$mol$\cdot$mol$^{-1}$) | (mmol$\cdot$m$^{-2}\cdot$s$^{-1}$) | (mol$\cdot$m$^{-2}\cdot$s$^{-1}$) | ($\mu$mol$\cdot$m$^{-2}\cdot$s$^{-1}$) |
| CK | 377.5 ± 8.53b | 4.08 ± 0.28b | 322 ± 29.17b | 5.53 ± 0.44b |
| Q | 421.5 ± 11.42a | 6.65 ± 0.36a | 531.5 ± 19.4a | 8.21 ± 0.59a |
| B | 381.5 ± 4.84b | 6.9 ± 0.59a | 485.75 ± 18.87a | 7.25 ± 0.55ab |
| D | 398.5 ± 4.52ab | 6.4 ± 0.15a | 524.25 ± 24.85a | 6.1 ± 0.66b |

CK: Continuous tomato cropping; Q: Celery/tomato rotation; B: Cabbage/tomato rotation; D: Kidney bean/tomato rotation; Ci: Intercellular $CO_2$ concentration; Tr: Transpiration rate; Gs: Stomatal conductance; Pn: Net photosynthetic rate. Different lowercase letters at each phenological stage indicate that the differences are statistically significant (P < 0.05).

## Effects of the different vegetable rotations on the fluorescence parameters of the tomato leaves

As shown in Table 3, there was no significant difference in Fv/Fm and Fv/Fo values between the leaves of each vegetable rotation and CK. The D leaves had the highest $q^P$ value, which was significantly higher (4%) than that of CK. However, there were no significant differences between the B, Q, and CK leaves for this parameter. The control CK leaves had the highest NPQ value, whereas the Q leaves had the lowest (47.83% lower than CK), and the NPQ value of B leaf was second only to that of Q leaf, which was significantly lower than that of the control by 43.48%. There was no significant difference in NPQ values between the D and CK leaves.

## Discussion

Bacteria are the major drivers of nearly all biogeochemical cycles in terrestrial ecosystems and participate in maintaining the health and productivity of soils in agricultural systems [28]. Shiomi et al. found that it is difficult for pathogenic bacteria to grow and reproduce in soil with a high bacterial diversity, owing to the generally antagonistic and competitive nature of bacteria [29]. Studies have shown that crop rotation provides an abundant diversity of bacterial groups in soil and that the most diverse rotations (four different crops) also have the most diverse and active soil microbiota [18]. Therefore, the negative effects of agricultural intensification can be mitigated by designing more specific and diverse crop rotations [30]. It has also been shown that the structure and diversity indices of the soil bacterial community do not change significantly under rotation [31]. In our study, bacterial alpha-diversity analysis revealed that the observed species of the B (cabbage) rotation were the highest, which was

**Table 3. Effects of different vegetable rotations on the fluorescence parameters of tomato leaves.**

| Cropping system | Fv/Fm | Fv/Fo | $q^P$ | NPQ |
|---|---|---|---|---|
| CK | 0.79 ± 0.01a | 3.28 ± 0.19a | 0.75 ± 0.01b | 0.23 ± 0.02a |
| Q | 0.82 ± 0.02a | 3.81 ± 0.29a | 0.76 ± 0.02ab | 0.12 ± 0.01b |
| B | 0.81 ± 0.01a | 3.72 ± 0.47a | 0.77 ± 0.01ab | 0.13 ± 0.01b |
| D | 0.8 ± 0.01a | 3.42 ± 0.3a | 0.78 ± 0.003a | 0.16 ± 0.02ab |

CK: Continuous tomato cropping; Q: Celery/tomato rotation; B: Cabbage/tomato rotation; D: Kidney bean/tomato rotation; Fv: Variable fluorescence; Fm: Maximum fluorescence; Fo: Initial fluorescence; $q^P$: Level of photochemical quenching of photosystem II; NPQ: Non-photochemical quenching. Different lowercase letters at each phenological stage indicate that the differences are statistically significant (P < 0.05).

significantly higher than CK; that is, the B rotation had the highest bacterial richness, followed by the Q (celery) rotation. Meanwhile, compared with CK, observed species, Shannon index and Chao1 index of all the three treatments including Q, B and D showed an increasing trend, indicating that crop rotation could improve the abundance and diversity of substrate bacteria. Therefore, compared with continuous monoculture, rotation can improve the diversity and richness of soil bacterial communities, as has been reported in many studies [14,32].

Among the soil microorganisms, bacteria generally account for the highest proportion, and changes in their community structure are affected by the planted crops. Different crop species have different soil microbial environments as well as different compositions and abundance of bacterial community structures [33,34]. In this study, bacterial community hierarchical clustering and PCoA showed that the control CK substrate was significantly separated from substrates used in the rotations of cabbage and celery, that is to say, there were significant differences in bacterial community structure between continuous cropping substrate and substrate after rotation of cabbage and celery.

At the phylum level, the dominant group in most bacterial communities is Proteobacteria, which is the main phylum of heterotrophic gram-negative bacteria in various ecosystems, including soil, the plant leaf surface, the atmosphere, sea water, and freshwater. These bacteria can induce symbiotic N fixation with plants. The phylum contains not only pathogens of animals and plants but also beneficial bacteria that can inhibit pathogenic bacteria [35,36]. In this study, the abundance of Proteus in all three vegetable rotation substrates was higher than that in the CK substrate. This result is supported by the fact that rotation can change soil nutrients by secreting different root substances, thus increasing the abundance of Proteus [37]. Additionally, compared with continuous tomato cropping, celery rotation significantly increased the abundance of the Acidobacteria, Chloroflexus, and Firmicutes. These bacterial communities are important components of soil microbiotas and play important roles in promoting the healthy growth of plants, circulating soil material, and building a stable ecological environment [38]. According to the bacterial thermogram of this study, the dominant bacteria in the rotation substrates belonged to the original control CK substrate, and the similarity was very high. The abundance of *Sphingomonas* and *Gemmatimonas* in the CK substrate was slightly higher than that in the rotation substrates. This result is corroborated by the study of Li et al. [39]. In the rotation substrates (B, D, Q), the abundance of actinomycetes, rhizobia, and unclassified anaeromycetes was significantly increased, with the highest abundance of actinomycetes and unclassified anaeromycetes observed in the Q substrate. Actinomycetes are a group of bacteria that can effectively inhibit the activity of soil-borne pathogens [40], and their abundance in soil can be increased by rotation [11]. Anaerobes, which are key groups in aquatic sediments and wetland soils [41,42], are closely related to the degradation of organic carbon and organic compounds [43,44].

In this study, the physicochemical properties and microbial flora of the CK substrate were changed by the rotation of different vegetables in the previous crop cycle, thus affecting the growth physiology of subsequent tomato plants. It was found that the height, stem diameter, and leaf area of subsequent tomato plants were basically the same throughout the growth period, and the changes among the different crop rotations were not significant. This may be because our rotation experiment has only been carried out for one year, so the effect on tomato growth is not obvious. However, these three parameters were higher overall for the tomato plants grown under each crop rotation system than for those grown under the continuous cropping system. In the study of Calonego et al. [45], it was also shown that Soybean grows more vigorously under rotation conditions.

Photosynthetic fluorescence is one of the most sensitive physiological processes affected when plants are subjected to external stress. Stress-induced damage of the photosynthetic

structure hinders photosynthetic electron transport and reduces photosystem activity, thereby affecting photosynthetic carbon assimilation in the plant [46]. With the increase in continuous cropping years, this cropping system leads to decreases in the root vigor, chlorophyll content, superoxide dismutase activity, Pn, Gs, Tr, yield, and biomass of tomato plants, whereas crop rotation could largely alleviate these harmful effects [47,48]. In our study, the photosynthetic and fluorescence parameters of tomato plants grown under the continuous cropping system were worse than those grown under each rotation system, where the celery rotation significantly increased the Ci, Tr, Gs, and Pn of subsequent tomato leaves, and the cabbage and bean rotations significantly increased the Tr and Gs of subsequent tomato leaves relative to the values for the control CK group. Bean rotation significantly increased the $q^P$ of subsequent tomato leaves, whereas celery rotation significantly decreased the NPQ of subsequent tomatoes. By contrast, celery rotation was more beneficial for improving the photosynthetic and fluorescence physiology of subsequent tomatoes. Ahmad et al. [49] also showed that the intercropping of garlic could significantly increase the chlorophyll content, Pn, and antioxidant enzyme activity of pepper, and the allelochemicals of garlic played a crucial role in improving the physiological properties and biochemical reactions of pepper. Celery, used in our study, also belongs to a class of vegetables with strong allergens, which again proves the importance of allergens in alleviating the substrate disorders caused by continuous cropping.

## Conclusion

In recent years, the Gobi agriculture gradually developed in Western China has developed and utilized a large area of non-arable hectares, boosts food production and enhances rural socioeconomics, and the cultivation substrate is an indispensable part of Gobi agriculture [50]. However, too long years of substrate cultivation will aggravate crop diseases and insect pests and appear continuous cropping obstacles. Replacing substrate requires a lot of manpower and material resources [51]. Rotation is a farming method, which can not only effectively alleviate the obstacles of continuous cropping, but also reduce the production cost of substrate cultivation. In this study, the effects of rotation of cabbage, kidney bean and celery on bacterial diversity, community structure of continuous tomato cropping rhizosphere and growth of subsequent tomato were studied. All four cropping systems tested could improve the richness and diversity of bacteria in the continuous cropping substrate to a certain extent. Among them, the structure of substrate bacterial community in celery rotation was significantly different from that in continuous cropping. Additionally, the celery rotation significantly increased the abundance of beneficial bacteria, such as *Actinobacteria_unclassified* and *Anaerolineaceae_unclassified*. And the rotation of celery can better improve the growth and physiological condition of tomato. In the future research, we will extend our findings to field experiments for verification, in order to apply the cultivation method of rotating celery to alleviate the obstacle of tomato continuous cropping in practical production.

## Supporting information

**S1 Fig. Effects of different vegetable rotations on tomato plant growth.** CK: Continuous tomato cropping; Q: Celery/tomato rotation; B: Cabbage/tomato rotation; D: Kidney bean/tomato rotation. Different lowercase letters at each phenological stage indicate that the differences are statistically significant (P < 0.05).
(DOCX)

**S1 Table. Comparison of species numbers and 16S (bacterial) diversity indices observed in the different cropping systems.** CK: Continuous tomato cropping; Q: Celery/tomato rotation;

B: Cabbage/tomato rotation; D: Kidney bean/tomato rotation. Different lowercase letters in a row indicate that the differences are statistically significant (P < 0.05). Observed species: Community richness; Shannon index: Community diversity; Simpson index: Community diversity; Chao1 index: Community richness.
(DOCX)

**S2 Table. Effects of different vegetable rotations on light and photosynthetic parameters of the tomato leaves.** CK: Continuous tomato cropping; Q: Celery/tomato rotation; B: Cabbage/tomato rotation; D: Kidney bean/tomato rotation; Ci: Intercellular CO2 concentration; Tr: Transpiration rate; Gs: Stomatal conductance; Pn: Net photosynthetic rate. Different lowercase letters at each phenological stage indicate that the differences are statistically significant (P < 0.05).
(DOCX)

**S3 Table. Effects of different vegetable rotations on the fluorescence parameters of tomato leaves.** CK: Continuous tomato cropping; Q: Celery/tomato rotation; B: Cabbage/tomato rotation; D: Kidney bean/tomato rotation; Fv: Variable fluorescence; Fm: Maximum fluorescence; Fo: Initial fluorescence; qp: Level of photochemical quenching of photosystem II; NPQ: Nonphotochemical quenching. Different lowercase letters at each phenological stage indicate that the differences are statistically significant (P < 0.05).
(DOCX)

## Acknowledgments

We sincerely thank the staff of the Vegetable Technical Service Center of Suzhou District, Jiuquan City, for experimental field management.

## Author Contributions

**Conceptualization:** Jian Lyu.

**Data curation:** Li Jin, Ning Jin.

**Formal analysis:** Li Jin.

**Funding acquisition:** Jian Lyu, Jihua Yu.

**Investigation:** Jian Lyu, Guobin Zhang, Zhi Feng.

**Methodology:** Jian Lyu.

**Project administration:** Jian Lyu, Jihua Yu.

**Resources:** Jianming Xie, Guobin Zhang, Zhi Feng, Jihua Yu.

**Software:** Zhongqi Tang, Zeci Liu.

**Supervision:** Jianming Xie, Shilei Luo, Jihua Yu.

**Validation:** Jianming Xie, Guobin Zhang.

**Visualization:** Li Jin, Ning Jin, Yue Wu, Zhongqi Tang, Zeci Liu.

**Writing – review & editing:** Yue Wu, Shilei Luo.

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
