## [Decision Letter · Decision Letter 0]

4 Aug 2021

PONE-D-21-19184

Effects of Different Vegetable Rotations on the rhizosphere Bacterial Community and Tomato Growth in a Continuous Tomato Cropping Substrate

PLOS ONE

Dear Dr. Jian,

Thank you for submitting your manuscript to PLOS ONE. After careful consideration, we feel that it has merit but does not fully meet PLOS ONE’s publication criteria as it currently stands. Therefore, we invite you to submit a revised version of the manuscript that addresses the points raised during the review process.

I specifically want you to modify the written part of the results of Table 1 in the result section as well as in the Discussion. Further, all necessary corrections pointed out by the reviewers should be corrected and updated with more recent and relevant references.  

We look forward to receiving your revised manuscript.

Kind regards,

Birinchi Sarma, PhD

Academic Editor

PLOS ONE

Journal Requirements:

Reviewers' comments:

Reviewer's Responses to Questions

**Comments to the Author**

1. Is the manuscript technically sound, and do the data support the conclusions?

Reviewer #1: Yes

Reviewer #2: Yes

2. Has the statistical analysis been performed appropriately and rigorously? 

Reviewer #1: Yes

Reviewer #2: Yes

3. Have the authors made all data underlying the findings in their manuscript fully available?

Reviewer #1: Yes

Reviewer #2: Yes

4. Is the manuscript presented in an intelligible fashion and written in standard English?

Reviewer #1: Yes

Reviewer #2: Yes

5. Review Comments to the Author

Reviewer #1: The authors performed a good and interesting work confirming, again, the advantages of three crop rotations (tomato/cabbage, tomato/kidney bean, and tomato/celery) vs a tomato monoculture in a horticultural soil-less system by using a potting soil under greenhouse condition with horticultural crops (I prefer this definition rather than “vegetables”) in China. They investigated more aspects regarding overall the soil quality of a potting mix substrate by correlating it with the productivity of a tomato crop by a multivariate approach. They have not nevertheless considered any crop rotation under field condition.

The paper is overall well written, scientifically sound, easily readable, and contains some novelty and originality traits. I recommend to accept after minor revision. Follow my comments and suggestions to improve it.

L.51. Please quote the sentence. I suggest to read the following: https://doi.org/10.1016/j.apsoil.2020.103601.

L.52-56. I suggest to expand this section with more references by describing all advantage and weakness of crop rotation between horticultural/herbaceous crops vs monoculture.

L.56. Please add upgraded references from the following paper Pervaiz, Z.H.; Iqbal, J.; Zhang, Q.; Chen, D.; Wei, H.; Saleem, M. Continuous Cropping Alters Multiple Biotic and Abiotic Indicators of Soil Health. Soil Syst. 2020, 4, 59.

L.68-69. Please give major details on organic ecotype soilless culture substrate used in this study (e.g. origin, spread and use into horticultural soil-less systems, etc.) by quoting them.

L.88-89. Please indicate the cultivar’ names of cabbage, kidney bean, and celery by giving major agronomical details of their cultivation.

L.136,139. Please delete “Determination of the…” from each sub-title.

L.327-328. Please quote the sentence: “Studies have shown that crop rotation provides an abundant diversity of bacterial groups in soil and that the most diverse rotations (four different crops) also have the most diverse and active soil microbiota (reference).”. I suggest to read the following: https://doi.org/10.1016/j.apsoil.2020.103601.

L.338-339. Please quote the sentence with upgraded references: “Therefore, compared with continuous monoculture, rotation can improve the diversity and richness of soil bacterial communities, as has been reported in many studies [26-28].”. I suggest to read the following reference: Pervaiz, Z.H.; Iqbal, J.; Zhang, Q.; Chen, D.; Wei, H.; Saleem, M. Continuous Cropping Alters Multiple Biotic and Abiotic Indicators of Soil Health. Soil Syst. 2020, 4, 59.

L.368. Please delete (Fig. 4).

L.369-370. Please quote the sentence: “The available P, total N, total K, organic matter, EC, and pH were the main factors affecting the soil bacterial community structure.” I suggest to read the following reference: Lavecchia, A.; Curci, M.; Jangid, K.; Whitman, W.B.; Ricciuti, P.; Pascazio, S.; Crecchio, C. Microbial 16S gene-based composition of a sorghum cropped rhizosphere soil under different fertilization managements. Biol. Fertil. Soils 2015, 51, 661–672.

L.413. Please rephrase as “In this study,….”.

Conclusions. The authors should address own future researches on this topic extending your findings into field condition because crop rotation is an agronomical practice more suitable in open field rather than in greenhouse. In horticultural soil-less systems, in fact, more else strategies are suitable than crop rotation to improve productivity and quality of tomato crop. This future perspective should be strongly highlighted in this section.

Figs.2,3. Please zoom the labels in the both axes, they are too little.

Fig.5. Please check all significance letters in the second box plot (Stem thickness vs Early flowering). I think there is a mistake.

Reviewer #2: My main observation is about the result shown in Table 1.

In Results section, I disagree with the authors, according to my opinion all these indices (Shannon, Simpson, Chao1 index), differences were not significant, however, an increased number of observed species occurred in B

(cabbage/tomato) compared to the CK (continuous tomato cropping). I consider that this paragraph should be rewritten and then modified in the discussion.

According to my experience, when the differences are not significant, I only can say that there is a trend and cannot order from highest to lowest.

I make many suggestions, in the attempt to help authors to prepare a better version of this manuscript.

In Fig. 2, I suggest increasing the letter size and put it in horizontal form.

In Fig.4, I recommend deleting g_ in each bacteria name to help to understand the analysis.

I suggest moving the percentage date. For Example, on lines 279-80

“At the beginning of flowering, the height of the Q plant was the highest, being significantly higher than that of the CK plant (by 0.44%)”.

Please, change by “At the beginning of flowering, the height of the Q plant was the highest (0.44%), being significantly higher than that of the CK plant.

- On lines 281-282: “At the early fruiting stage, the D plant was significantly taller (0.46%), than the CK plant. And so on with the next…

The results show that these growth parameters are slightly modified. Please consider it in the discussion section.

In the M&M section, I consider that the sentence “The materials and methods are similar to those of “Lyu” [2]:” maybe incorporated inside experimental design.

6. PLOS authors have the option to publish the peer review history of their article (what does this mean?). If published, this will include your full peer review and any attached files.

Reviewer #1: No

Reviewer #2: No

---

## [Author Response · Author response to Decision Letter 0]

21 Aug 2021

Response to Comments of Reviewer#1:

Q1. L.51. Please quote the sentence. I suggest to read the following: https://doi.org/10.1016/j.apsoil.2020.103601.

Response: Thank you for your kind suggestion. We have quoted the reference to the manuscript. (Line59)

Q2. L.52-56. I suggest to expand this section with more references by describing all advantage and weakness of crop rotation between horticultural/herbaceous crops vs monoculture.

Response: Thank you for your kind comment. We have supplemented all advantage and weakness of crop rotation between horticultural/herbaceous crops vs monoculture in the manuscript. The corresponding sentences in the manuscript were revised as follows: “It is well known that the continuous cropping of a single species causes plant growth retardation, serious diseases and insect pests, low crop productivity, soil degradation, and microecosystem imbalance [6, 7]. Compared with monoculture，Crop rotation (e.g., leguminous crop diversification) is a strategy used for maintaining soil quality and crop productivity to the high degree that single cropping cannot [8]. Horticultural/herbaceous crop rotation affects the physical and chemical properties of soil and its microbial community through the difference in the quantity and composition of root exudates, so as to improve the soil biological environment, reduce continuous cropping obstacles, and improve the yield of horticultural crops. Moreover, rotation can maintain soil properties within an acceptable range and achieve the goal of sustainable agriculture [9-12]. Additionally, the different plant–microbe interactions also affect the soil microbial communities, leading to less soil-borne diseases [13, 14]. The diversity and richness of soil microorganisms play a particularly important role in maintaining soil health and promoting crop growth [15]. In recent years, most studies on ways to overcome or alleviate the obstacles of continuous cropping have focused mainly on changes in the rhizosphere soil microorganisms after the continuous cultivation of crops [16, 17]. However, there are few reports on the changes in the organic substrate quality and microorganisms effect by tomato monoculture for several years [18].” (Line44-59)

Q3. L.56. Please add upgraded references from the following paper Pervaiz, Z.H.; Iqbal, J.; Zhang, Q.; Chen, D.; Wei, H.; Saleem, M. Continuous Cropping Alters Multiple Biotic and Abiotic Indicators of Soil Health. Soil Syst. 2020, 4, 59.

Response: Thank you for your kind comment. We have added the updated references to the manuscript. (Line54)

Q4. L.68-69. Please give major details on organic ecotype soilless culture substrate used in this study (e.g. origin, spread and use into horticultural soil-less systems, etc.) by quoting them.

Response: Thank you for your kind comment. We have supplemented the main details of the organic ecotype soilless culture substrate in the manuscript and added corresponding references. The corresponding sentence is as follows: “The test substrate is an organic ecotype soilless culture substrate derived from local agricultural waste, which is made up of a mixture of slag, spent mushroom, cow manure, chicken manure, and corn straw at the ratios of 13:5:5:2:14, respectively. It is mainly suitable for tomato substrate cultivation and has been vigorously promoted locally [19].” (Line71-74)

Q5. L.88-89. Please indicate the cultivar’ names of cabbage, kidney bean, and celery by giving major agronomical details of their cultivation.

Response: Thank you for your kind suggestion. We have added the variety names of cabbage, kidney bean and celery to the manuscript, and supplemented the main cultivation details. The corresponding sentences were as follows: “Four cabbage plants are planted in each pot, a small amount of water is watered for many times in the early stage, and watering and fertilization are applied once a week in the peak growth period. The growth cycle of cabbage is short, so a total of two crops are planted. Two kidney beans are planted in each pot, and the vine is hung during vine extension. Fertilization and irrigation are carried out every two weeks in the early growth stage, and once every week in the peak pod setting stage. Two celery plants are planted in each pot. After planting, a small amount of water is watered for many times to promote the growth of seedlings. There is less water in the early stage of growth to prevent overgrowth. When entering the period of vigorous nutritional growth, water and fertilizer are applied once a week. Other field management measures were consistent with local conventional management measures.” (Line94-95, Line99-108)

Q6. L.136,139. Please delete “Determination of the…” from each sub-title.

Response: Thank you for your kind comment. We have deleted " Determination of the…" from the manuscript (Line148,151)

Q7. L.327-328. Please quote the sentence: “Studies have shown that crop rotation provides an abundant diversity of bacterial groups in soil and that the most diverse rotations (four different crops) also have the most diverse and active soil microbiota (reference).”. I suggest to read the following: https://doi.org/10.1016/j.apsoil.2020.103601.

Response: Thank you for your kind comment. We have cited the references you provided in the manuscript. (Line298-300)

Q8. L.338-339. Please quote the sentence with upgraded references: “Therefore, compared with continuous monoculture, rotation can improve the diversity and richness of soil bacterial communities, as has been reported in many studies [26-28].”. I suggest to read the following reference: Pervaiz, Z.H.; Iqbal, J.; Zhang, Q.; Chen, D.; Wei, H.; Saleem, M. Continuous Cropping Alters Multiple Biotic and Abiotic Indicators of Soil Health. Soil Syst. 2020, 4, 59.

Response: Thank you for your kind comment. We have quoted the updated references into the manuscript. (Line308-310)

Q9. L.368. Please delete (Fig. 4).

Response: Thank you for your kind comment. We have deleted Fig. 4 from the manuscript.

Q10. L.369-370. Please quote the sentence: “The available P, total N, total K, organic matter, EC, and pH were the main factors affecting the soil bacterial community structure.” I suggest to read the following reference: Lavecchia, A.; Curci, M.; Jangid, K.; Whitman, W.B.; Ricciuti, P.; Pascazio, S.; Crecchio, C. Microbial 16S gene-based composition of a sorghum cropped rhizosphere soil under different fertilization managements. Biol. Fertil. Soils 2015, 51, 661–672.

Response: Thank you for your kind comment. Because I have deleted Fig. 4 from the result, I have also deleted the corresponding discussion part from the manuscript.

Q11. L.413. Please rephrase as “In this study, ….”

Response: Thank you for your kind comment. We have corrected the corresponding sentences in the manuscript. (Line374)

Q12. Conclusions. The authors should address own future researches on this topic extending your findings into field condition because crop rotation is an agronomical practice more suitable in open field rather than in greenhouse. In horticultural soil-less systems, in fact, more else strategies are suitable than crop rotation to improve productivity and quality of tomato crop. This future perspective should be strongly highlighted in this section.

Response: Thank you for your kind suggestion. We have rewritten the conclusion, and the corresponding sentences are as follows: “In recent years, the Gobi agriculture gradually developed in Western China has developed and utilized a large area of non-arable hectares, boosts food production and enhances rural socioeconomics, and the cultivation substrate is an indispensable part of Gobi agriculture [53]. However, too long years of substrate cultivation will aggravate crop diseases and insect pests and appear continuous cropping obstacles. Replacing substrate requires a lot of manpower and material resources [54]. Rotation is a farming method, which can not only effectively alleviate the obstacles of continuous cropping, but also reduce the production cost of substrate cultivation. In this study, the effects of rotation of cabbage, kidney bean and celery on bacterial diversity, community structure of continuous tomato cropping rhizosphere and growth of subsequent tomato were studied. All four cropping systems tested could improve the richness and diversity of bacteria in the continuous cropping substrate to a certain extent. Among them, the structure of substrate bacterial community in celery rotation was significantly different from that in continuous cropping. Additionally, the celery rotation significantly increased the abundance of beneficial bacteria, such as Actinobacteria_unclassified and Anaerolineaceae_unclassified. And the rotation of celery can better improve the growth and physiological condition of tomato. In the future research, we will extend our findings to field experiments for verification, in order to apply the cultivation method of rotating celery to alleviate the obstacle of tomato continuous cropping in practical production.” (Line 368-384)

Q13. Figs.2,3. Please zoom the labels in the both axes, they are too little.

Response: Thank you for your kind suggestion. We have scaled the the labels in the both axes of figures 2 and 3, and the corresponding modifications are as follows:

Fig.2 

Fig.3

Q14. Fig.5. Please check all significance letters in the second box plot (Stem thickness vs Early flowering). I think there is a mistake

Response: Thank you for your kind suggestion. We have corrected the letters in Figure 5, and the corresponding modifications are as follows:

Fig.5

Response to Comments of Reviewer # 2

Q1. My main observation is about the result shown in Table 1.

In Results section, I disagree with the authors, according to my opinion all these indices (Shannon, Simpson, Chao1 index), differences were not significant, however, an increased number of observed species occurred in B (cabbage/tomato) compared to the CK (continuous tomato cropping). I consider that this paragraph should be rewritten and then modified in the discussion.

According to my experience, when the differences are not significant, I only can say that there is a trend and cannot order from highest to lowest.

Response: Thank you for your kind comment. For table 1, we have made corresponding modifications in the results and discussion. The corresponding sentences in the manuscript are as follows:

Result：“Additionally, compared with CK, the 3 vegetable rotations showed an increasing trend for the observed species 、Shannon index and Chao1 index of bacteria.”(Line180-181)

Discussion: “In our study, bacterial alpha-diversity analysis revealed that the observed species of the B (cabbage) rotation were the highest, which was significantly higher than CK; that is, the B rotation had the highest bacterial richness, followed by the Q (celery) rotation. Meanwhile, compared with CK, observed species, Shannon index and Chao1 index of all the three treatments including Q, B and D showed an increasing trend, indicating that crop rotation could improve the abundance and diversity of substrate bacteria. Therefore, compared with continuous monoculture, rotation can improve the diversity and richness of soil bacterial communities, as has been reported in many studies.” (Line303-310)

Q2. In Fig. 2, I suggest increasing the letter size and put it in horizontal form.

Response: Thank you for your kind suggestion. We have increased the letter size in Figure 2 and put it in horizontal form. The modified Figure 2 is as follows:

Q3. In Fig.4, I recommend deleting g_ in each bacteria name to help to understand the analysis.

Response: Thank you for your kind comment. After our consideration, we decided to delete Fig. 4 from the manuscript. Because reviewer #1 suggested deleting Fig. 4. We also consulted the editor again, and the editor also thought that Fig. 4 was of little significance. In addition, the deletion of Fig. 4 does not affect the conclusion of the full text. Therefore, we deleted Fig. 4 from the manuscript. Thank you again for your valuable comments.

 Q4. I suggest moving the percentage date. For Example, on lines 279-80

“At the beginning of flowering, the height of the Q plant was the highest, being significantly higher than that of the CK plant (by 0.44%)”.

Please, change by “At the beginning of flowering, the height of the Q plant was the highest (0.44%), being significantly higher than that of the CK plant.

Response: Thank you for your helpful suggestion. We have changed by “At the beginning of flowering, the height of the Q plant was the highest (0.44%), being significantly higher than that of the CK plant.” (Line250)

Q5. On lines 281-282: “At the early fruiting stage, the D plant was significantly taller (0.46%), than the CK plant. And so on with the next…

The results show that these growth parameters are slightly modified. Please consider it in the discussion section.

Response: Thank you for your helpful suggestion. We have moved all the percentage date in the manuscript to the appropriate position. (Line252,261). In the discussion part, we further discussed the result of " these growth parameters are slightly modified ", and the corresponding sentences are as follows: “It was found that the height, stem diameter, and leaf area of subsequent tomato plants were basically the same throughout the growth period, and the changes among the different crop rotations were not significant. This may be because our rotation experiment has only been carried out for one year, so the effect on tomato growth is not obvious.” (Line341-344)

Q6. In the M&M section, I consider that the sentence “The materials and methods are similar to those of “Lyu” [2]:” maybe incorporated inside experimental design.

Response: Thank you for your kind comment. We have deleted " The materials and methods are similar to those of “Lyu” [2]: " in materials and methods and added " The Experimental design is similar to those of “Lyu” [2]: " in experimental design. (Line70)

---

## [Editor Report · Decision Letter 1]

1 Sep 2021

Effects of Different Vegetable Rotations on the rhizosphere Bacterial Community and Tomato Growth in a Continuous Tomato Cropping Substrate

PONE-D-21-19184R1

Dear Dr. Jian,

We’re pleased to inform you that your manuscript has been judged scientifically suitable for publication and will be formally accepted for publication once it meets all outstanding technical requirements.

Kind regards,

Birinchi Sarma, PhD

Academic Editor

PLOS ONE
---

## [Editor Report · Acceptance letter]

14 Sep 2021

PONE-D-21-19184R1 

Effects of different vegetable rotations on the rhizosphere bacterial community and tomato growth in a continuous tomato cropping substrate 

Dear Dr. Lyu:

I'm pleased to inform you that your manuscript has been deemed suitable for publication in PLOS ONE. Congratulations! Your manuscript is now with our production department. 

Kind regards, 

on behalf of

Dr. Birinchi Sarma 

Academic Editor

PLOS ONE